

# Influence of foetal inflammation on the development of meconium aspiration syndrome in term neonates with meconium-stained amniotic fluid

Kyoko Yokoi[1], Osuke Iwata[2], Satoru Kobayashi[1], Kanji Muramatsu[1] and Haruo Goto[1]

[1] Department of Pediatrics, Nagoya West Medical Center, Nagoya, Japan
[2] Department of Neonatology and Pediatrics, Nagoya City University Graduate School of Medical Science, Nagoya, Japan

Corresponding author
Kyoko Yokoi,
kyoyo0410@yahoo.co.jp

## ABSTRACT

**Background:** Meconium-stained amniotic fluid is observed in approximately 10–15% of all deliveries; however, only 5% of neonates with meconium-stained amniotic fluid develop meconium aspiration syndrome (MAS). Although foetal distress and subsequent sympathetic stimulation have been considered as the primary upstream events of MAS, this clinical complication sometimes occurs due to other pathologies, such as intraamniotic inflammation. The aim of this study was to investigate whether the incidence of MAS is associated with the presence of funisitis and chorioamnionitis in term neonates with meconium-stained amniotic fluid.
**Methods:** Between April 2013 and March 2015, a total of 95 term neonates with meconium-stained amniotic fluid, who were hospitalized at a neonatal intensive care unit, were enrolled in the study. The placenta and umbilical cord were histopathologically examined. Clinical variables and histopathological findings associated with the incidence of MAS were studied.
**Results:** A total of 36 neonates developed MAS. Univariate logistic regression analysis revealed that a heavier birth weight, male sex, 1-min Apgar score $\leq$ 7, funisitis (but not chorioamnionitis), and elevated acute-phase inflammatory reaction score were associated with increased incidence of MAS (all $p < 0.05$). The multivariate model comprised funisitis (OR = 5.03, 95% CI [1.63–15.5], 1-min Apgar score $\leq$ 7 (OR = 2.74, 95% CI [1.06–7.09], and male sex (OR = 3.4, 95% CI [1.24–9.34].
**Conclusion:** In neonates with meconium-stained amniotic fluid, funisitis, as well as low 1-min Apgar score and male sex, was identified as an independent variable for MAS development. Intraamniotic inflammation might be involved in the pathological mechanisms of MAS.

## INTRODUCTION

Meconium-stained amniotic fluid is observed in approximately 10–15% of all deliveries (*Yoder et al., 2002*), which has been considered as the primary cause of meconium

aspiration syndrome (MAS). However, only 5% of neonates with meconium-stained amniotic fluid develop MAS (*Yoder et al., 2002*; *Dargaville, Copnell & Australian and New Zealand Neonatal Network, 2006*; *Swarnam, Soraisham & Sivanandan, 2012*). In addition, the use of surfactant replacement, nitric oxide therapy and extracorporeal membrane oxygenation has led to a marked decrease in mortality and morbidity associated with MAS in the recent times (*Swarnam, Soraisham & Sivanandan, 2012*). Nevertheless, these therapeutic options are usually unavailable for neonates born in a low-resource setting. Even in a high-resource setting, advanced intervention is often delayed, because the current diagnostic algorithms of MAS based on clinical symptoms and X-ray patterns do not provide sufficient insight as to who are likely to develop severe respiratory failure (*Ahanya et al., 2005*; *Swarnam, Soraisham & Sivanandan, 2012*).

To establish early, reliable biomarkers of MAS, precise understanding of the pathological mechanism is essential. Although asphyxia and subsequent sympathetic stimulation have been considered to trigger foetal passage of meconium into amniotic fluid (*Ahanya et al., 2005*; *Van Ierland & De Beaufort, 2009*; *Swarnam, Soraisham & Sivanandan, 2012*; *Lindenskov et al., 2015*), MAS sometimes develops in the absence of overt episodes suggestive of antenatal asphyxia; several previous studies did not find an association between low Apgar scores, umbilical arterial acidosis, and incidence of MAS (*Trimmer & Gilstrap, 1991*; *Ramin et al., 1993*; *Blackwell et al., 2001*; *Oyelese et al., 2006*), suggesting the presence of additional triggers of MAS, such as intraamniotic inflammation.

Studies suggested that meconium-stained amniotic fluid is more likely to be contaminated with bacteria, endotoxin, and inflammatory mediators than clear amniotic fluid, and often accompanied by clinical evidence of intraamniotic inflammation (*Romero et al., 2014*). Meconium-stained amniotic fluid is also linked to findings suggestive of intraamniotic inflammation, which is represented by funisitis and chorioamnionitis (*Burgess & Hutchins, 1996*; *Romero et al., 2014*; *Choi et al., 2015*; *Lee et al., 2016*). While chorioamnionitis is suggestive of maternal inflammatory response, funisitis is considered to represent foetal inflammatory response (*Redline et al., 2003*). In preterm neonates with funisitis, elevated plasma interleukin-6 (IL-6) level in cord blood (*Yoon et al., 2000a*) is associated with increased incidence of adverse neonatal outcomes such as chronic lung disease, intracranial haemorrhage, and cerebral palsy (*Kim et al., 2001*; *Yoon et al., 2000b*). However, few studies in term neonates have investigated the influence of intraamniotic inflammation to the development of MAS.

We performed a prospective observational study to investigate whether the incidence of MAS is associated with the presence of funisitis and chorioamnionitis.

# METHODS

## Patients and methods

This study was conducted under the approval of the Ethics Committee of Nagoya West Medical Center (approval reference number: 18-04-315-07). The internal review board advised that there is no statutory requirement for parental consent for the use of clinical and histopathological data, which are collected as a routine clinical care in the unit.

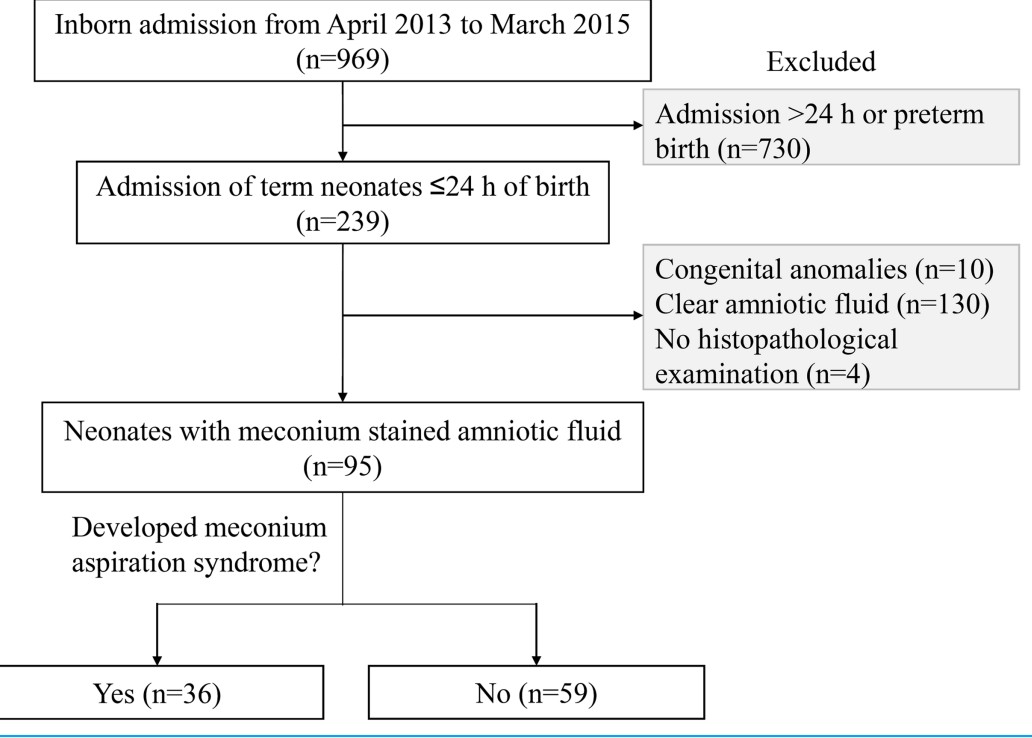

**Figure 1 Profile of the study population.** A total of 95 neonates, who were born through meconium-stained amniotic fluid and admitted to the neonatal intensive care unit within 24 h of birth, were studied.

## Study population

Between April 2013 and March 2015, 969 inborn neonates were hospitalized at a level-II neonatal intensive care unit (NICU) of Nagoya West Medical Center (Nagoya, Aichi, Japan), which provides intensive care for virtually all sick neonates, except for those who require advanced life support treatments, such as haemodialysis/filtration, cardiac surgery, and extracorporeal membrane oxygenation. Of these, 239 inborn neonates ≥37 weeks of gestation were admitted within 24 h of birth, whose blood samples on day 0 were available. Subsequently, 99 neonates were enrolled in the study after excluding 10 neonates with major congenital anomalies and 130 neonates without meconium-stained amniotic fluid (Fig. 1).

## Clinical variables

Presence of meconium-stained amniotic fluid was evaluated at birth under visual observation as previously described (*Lee et al., 2016*). MAS was defined as respiratory distress in neonates though meconium-stained amniotic fluid whose symptoms cannot be otherwise explained, requiring assisted mechanical ventilation or oxygen at a concentration of ≥40% for at least 48 h, and radiographic findings were consistent with MAS (*Fraser et al., 2005*; *Lee et al., 2016*). X-ray photographs were reviewed by an experienced staff neonatologist, and the placenta and umbilical cord were histopathologically examined by a single pathologist. Chorioamnionitis was defined as

infiltration of neutrophils identified in chorionic membranes (Stage II by Blanc's classification), whereas funisitis was defined as infiltration of neutrophils within the walls of umbilical vessels or in Wharton's jelly (*Blanc, 1979*; *Redline et al., 2003*). Other clinical variables were collected from patients' clinical record, including gestational age, birth weight and its standard score calculated against the standard Japanese birth weight for parity and gestational age (*Itabashi et al., 2014*), sex, delivery mode, premature rupture of the membranes, Apgar score at 1 and 5 min, cord blood pH, and the duration of invasive/non-invasive positive-pressure ventilation and oxygen supplementation.

## Laboratory studies

In our unit, at the admission of neonates born through meconium-stained amniotic fluid, routine blood tests are performed including blood gas analysis, cell counts, blood sugar level, and serum inflammatory markers. For the inflammatory markers, C-reactive protein (CRP), $\alpha_1$-acid glycoprotein ($\alpha_1AG$), and haptoglobin were measured using turbidimetric immunoassay (Quick Turbo; Shino-Test Corporation, Tokyo, Japan). CRP > 0.3 mg/dl, $\alpha_1AG$ > 20 mg/dl, and haptoglobin > 13 mg/dl were regarded as positive (*Speer, Bruns & Gahr, 1983*; *Ipek, Saracoglu & Bozaykut, 2010*; *Nakamura et al., 2015*); the acute-phase inflammatory reaction scores of 0 (none positive), 1 (one positive biomarker), 2 (two positive biomarkers), and 3 (all three positive biomarkers) were given based on the number of positive inflammatory biomarkers (*Nakamura et al., 2015*). Venous blood culture was assessed for neonates with clinical signs suggestive of antenatal infection (e.g., elevation of maternal white blood cell count and CRP, maternal pyrexia, and premature rupture of the membranes) or with the acute-phase inflammatory reaction scores $\geq 1$.

## Data analysis

The clinical variables were compared between neonates with and without MAS. Apgar scores ($\leq 7$) and cord blood pH (<7.1) were divided using clinically relevant thresholds. All statistical analyses were performed using R (The R Foundation for Statistical Computing, Vienna, Austria, version 1.22) and its Japanese interface (EZR; Saitama Medical Center, Jichi Medical University, Saitama, Japan) (*Kanda, 2013*). Logistic regression analyses were performed to develop a model explaining MAS development using funisitis and chorioamnionitis as mandatory independent variables (only one of the two mandatory variables were tested at a time because of the collinearity observed between each other). Statistical findings were not corrected for multiple comparisons for univariate analysis, because of the exploratory nature of this study.

## RESULTS

A total of 99 term neonates with meconium-stained amniotic fluid, histopathological examination was not available in four neonates, whose data were not considered further. Of 95 remaining neonates, 36 neonates (37.9%) developed MAS, none of whom showed positive blood culture. Gestational age, caesarean delivery, and incidence of premature rupture of the membranes and cord blood acidosis did not differ between neonates who developed and did not develop MAS (Table 1). Univariate logistic regression analysis

**Table 1 Independent variables of meconium aspiration syndrome development.**

| | Meconium aspiration syndrome | | Odds ratio | | | p-value |
|---|---|---|---|---|---|---|
| | No (n = 59) | Yes (n = 36) | Mean | 95% Cl | | |
| | | | | Lower | Upper | |
| **Univariate model** | | | | | | |
| Gestation (weeks) | 40.3 ± 0.3 | 40.5 ± 0.4 | 1.11 | 0.74 | 1.66 | 0.610 |
| Birth weight (kg) | 3.02 ± 0.56 | 3.24 ± 0.45 | 2.38 | 1.00 | 5.66 | 0.050 |
| Birth weight z-score | −0.32 ± 1.48 | 0.21 ± 1.25 | 1.32 | 0.97 | 1.81 | 0.080 |
| SGA < 2 SD of norm | 11 (19) | 2 (6) | 0.26 | 0.03 | 1.31 | 0.259 |
| Male sex | 31 (54) | 28 (78) | 3.16 | 1.24 | 8.07 | 0.016 |
| Delivery mode | | | 1.45 | 0.89 | 2.34 | 0.136 |
| Vaginal | 34 (58) | 16 (44) | 0.56 | 0.24 | 1.29 | 0.171 |
| Forceps/vacuum | 11 (19) | 7 (19) | 0.91 | 0.32 | 2.58 | 0.865 |
| Caesarean | 12 (20) | 13 (36) | 2.15 | 0.87 | 5.33 | 0.097 |
| PROM > 24 h | 13 (22) | 11 (31) | 1.56 | 0.61 | 3.98 | 0.355 |
| 1-min Apgar ≤ 7 | 20 (34) | 21 (58) | 2.44 | 1.04 | 5.70 | 0.040 |
| 5-min Apgar ≤ 7 | 12 (20) | 9 (25) | 1.12 | 0.41 | 3.07 | 0.827 |
| Cord blood pH < 7.1 | 7 (12) | 5 (14) | 1.20 | 0.35 | 4.10 | 0.773 |
| Cord blood pH < 7.2 | 22 (37) | 19 (53) | 1.88 | 0.81 | 4.36 | 0.141 |
| Chorioamnionitis | 26 (44) | 20 (56) | 1.59 | 0.69 | 3.65 | 0.278 |
| Funisitis | 35 (59) | 30 (83) | 3.43 | 1.24 | 9.50 | 0.018 |
| Inflammatory biomarkers at admission | | | | | | |
| CRP (>0.3 mg/dl) | 8 (14) | 9 (25) | 1.18 | 0.83 | 1.69 | 0.354 |
| α1-AG (>20 mg/dl) | 38 (64) | 28 (78) | 1.01 | 1.00 | 1.03 | 0.101 |
| HP (>13 mg/dl) | 8 (14) | 10 (28) | 1.02 | 1.00 | 1.04 | 0.060 |
| Acute-phase inflammatory reaction score | | | | | | |
| 0 | 21 (36) | 8 (22) | Reference | | | |
| 1 or 2 | 33 (55) | 19 (53) | 1.59 | 0.59 | 4.26 | 0.356 |
| 3 | 5 (9) | 9 (25) | 4.72 | 1.21 | 18.5 | 0.026 |
| **Multivariate model** | | | | | | |
| Funisitis | | | 5.03 | 1.63 | 15.5 | 0.005 |
| 1-min Apgar ≤ 7 | | | 2.74 | 1.06 | 7.09 | 0.038 |
| Male sex | | | 3.4 | 1.24 | 9.34 | 0.018 |

**Notes:**
Values are shown as mean ± standard deviation or number (%).
CI, confidence interval; CRP, C-reactive protein; $\alpha_1$AG, $\alpha_1$-acid glycoprotein; HP, haptoglobin; SD, standard deviation; SGA, small for gestational age.

showed that a high birth weight, male sex, 1-min Apgar score ≤ 7, funisitis, and a high acute-phase inflammatory reaction score of 3 (compared with 0) were associated with increased incidence of MAS (all $p < 0.05$; Table 1). The final model to explain the incidence of MAS in neonates with meconium-stained amniotic fluid comprised funisitis (OR = 5.03; 95% CI [1.63–15.5], low 1-min Apgar score ≤ 7 (OR = 2.74; 95% CI [1.06–7.09], and male sex (OR = 3.4; 95% CI [1.24–9.34], where chorioamnionitis was not identified as an independent variable to explain MAS development (See Table S1 for the multivariate

model, which used chorioamnionitis instead of funisitis; and Tables S2 and S3 for the multivariate model, which used the duration of mechanical ventilation and oxygen supplementation as alternative dependent variables).

## DISCUSSION

Meconium aspiration syndrome is one of the most commonly recognized respiratory diseases in neonates, however, the exact pathological mechanism leading to foetal meconium passage into the amniotic fluid and its aspiration is still unknown. Our study suggested that, in neonates with meconium-stained amniotic fluid, funisitis, as well as low 1-min Apgar scores and male sex, was associated with increased incidence of MAS, pointing toward the role of inflammatory factors in the pathophysiology of MAS. Further large-scale studies are required to delineate the role of antenatal inflammation in the development of meconium-stained amniotic fluid and MAS.

Previous studies highlighted that MAS development is associated with low cord blood pH and Apgar scores at 5 min (*Dargaville, Copnell & Australian and New Zealand Neonatal Network, 2006*; *Van Ierland & De Beaufort, 2009*; *Karabayir, Demirel & Bayramoglu, 2015*), supporting the hypothesis that hypoxic-ischemic events trigger foetal meconium passage into the amniotic fluid and its subsequent aspiration. However, several studies failed to demonstrate the association between cord blood pH, Apgar scores, and MAS development (*Trimmer & Gilstrap, 1991*; *Ramin et al., 1993*; *Blackwell et al., 2001*; *Oyelese et al., 2006*). Our study also found that only the 1-min Apgar scores, but not the 5-min Apgar scores, and cord blood pH, were associated with MAS development, suggesting that antenatal hypoxic-ischemia is not sufficient to explain MAS development, and other control variables of MAS might exist.

In our study, funisitis was associated with an increased incidence of MAS, supporting a hypothesis that intraamniotic inflammation might be an additional factor involved in the development of MAS. A study that investigated autopsy cases with histopathologic evidence of intrauterine meconium exposure found that pulmonary inflammation was associated with inflammation of the umbilical cord, as well as meconium aspiration (*Romero et al., 2014*). *Thureen et al. (1997)* showed that three of six placentas with MAS showed funisitis associated with villus ischemic changes, which may have contributed to in utero foetal hypoxia. *Lee et al. (2016)* showed that neonates diagnosed with funisitis were at more than fourfold risks of developing MAS than those without funisitis, whereas *Kim, Oh & Kim (2017)* observed that the incidence and severity of funisitis were associated with those of MAS, suggesting that intraamniotic inflammation with foetal systemic inflammation might be an important upstream control variable in MAS. By performing placental histopathological examination virtually for all newborn intensive care unit (NICU) neonates, our study provided additional evidence to support the impact of intraamniotic inflammation to MAS development.

In our study, MAS development was associated with funisitis, but not chorioamnionitis. It is possible that chorioamnionitis might represent a maternal inflammatory response to intraamniotic infection, whereas funisitis might be predominantly represent the foetal inflammatory response, which is accompanied by elevation of inflammatory cytokine and

procytokine levels. Intraamniotic inflammation occurs more frequently in the presence of meconium-stained amniotic fluid than that in clear amniotic fluid, as shown by elevated amniotic fluid IL-6 levels in neonates with meconium-stained amniotic fluid (*Romero et al., 2014*). In animal models, foetal swallowing of amniotic fluid containing bacteria, endotoxin, and proinflammatory mediator is suggested to evoke inflammation and subsequent bowel peristalsis and meconium passage, which may explain both meconium passage in utero and its aspiration in association with foetal infection and inflammation. A clinical study confirmed that meconium with bacteria and inflammatory mediators may be aspirated in utero following stressful events with or without acidaemia. It is possible to extrapolate that intraamniotic inflammation triggers a range of foetal inflammatory response, which is represented by funisitis and elevation of inflammatory biomarkers. Foetal ingestion of amniotic fluid containing inflammatory mediators may ultimately induce the meconium passage into the amniotic fluid. Although our study did not measure inflammatory cytokine levels, three inflammatory biomarkers, CRP, $\alpha_1$AG, and haptoglobin, were employed, which are produced in the liver in response to intrinsic/extrinsic stimuli and subsequent elevation of IL-1ß and IL-6 (*Ipek, Saracoglu & Bozaykut, 2010*; *Nakamura et al., 2015*). Although these inflammatory biomarkers and their composite scores were not involved within the final model to explain the development of MAS, there was a consistent trend that an increase in inflammatory biomarkers was associated with relatively higher incidence of MAS. Consistent to our findings, *Hofer et al. (2016)* reported that elevated CRP levels and changes in white blood cell/neutrophil counts were associated with the severity of MAS, although this study simultaneously highlighted the potential impact of mechanical ventilation on the inflammatory biomarkers. Further accumulation of knowledge regarding the role of inflammatory reactions may accelerate the establishment of biomarkers, which provide early, precise prediction of MAS development in neonates, who are born through meconium-stained amniotic fluid.

It is known that inflammatory responses are sex-dependent; for example, studies in vivo demonstrated greater release of proinflammatory cytokines after in vivo lipopolysaccharide stimulation in male foetuses than in female foetuses (*Kim-Fine et al., 2012*; *Koch et al., 2014*). *Lambermont et al. (2012)* reported increased white blood count response in bronchoalveolar fluid and worse lung compliance following exposure to chorioamnionitis in male than in female neonates. In our study, male neonates had at a higher risk of developing MAS than female neonates. However, the size of our study population was too small to confirm the role of sex on MAS development.

Several limitations of our study were noted, which need to be considered when interpreting our findings. Although we performed standard histopathological examinations of the placenta and umbilical cord for all participating neonates, the study population is relatively small, which did not allow the assessment of some important clinical cofounders. To minimize the population bias, our study was conducted at a single level-II unit recruiting only inborn neonates. However, we were not able to include neonates, who did not have meconium-stained amniotic fluid, and those, who were not hospitalized at NICU, leaving uncertainly around the role of intraamniotic inflammation in the entire birth cohort. Several important clinical variables associated with antenatal inflammation

and hypoxic-ischemic stress, such as maternal fever, infection and finding from cardiotocography, were not incorporated within the analysis, leading to a difficulty in discriminating acute chorioamnionitis and funisitis from chronic ones. A prospective cohort study is currently underway to elucidate the role of intraamniotic inflammation on MAS development in all neonates born at our hospital by involving extensive maternal and neonatal variables.

## CONCLUSIONS

The incidence of MAS was associated with funisitis, as well as low 1-min Apgar scores and male sex, suggesting that intraamniotic inflammation might also be involved in the pathological mechanisms of MAS. Further large-scale prospective studies are needed to delineate the detailed mechanism of MAS and allow its efficient prediction and prevention.

## ACKNOWLEDGEMENTS

The authors are grateful to all clinical staffs of the Division of Obstetrics and Gynecology and the Division of Clinical Pathology, Nagoya West Medical Center, for their technical input in the histopathological examination of the placenta.

### Funding

The authors received no funding for this work.

### Competing Interests

The authors declare that they have no competing interests.

### Author Contributions

- Kyoko Yokoi conceived and designed the experiments, performed the experiments, analyzed the data, contributed reagents/materials/analysis tools.
- Osuke Iwata analyzed the data, contributed reagents/materials/analysis tools, prepared figures and/or tables.
- Satoru Kobayashi prepared figures and/or tables.
- Kanji Muramatsu collected clinical deta, and participated liturature research.
- Haruo Goto authored or reviewed drafts of the paper, approved the final draft.

### Ethics

The following information was supplied relating to ethical approvals (i.e., approving body and any reference numbers):

This study was conducted under the approval of the Ethics Committee of Nagoya West Medical Center (18-04-315-07).

### Data Availability

Raw data is available in the Supplemental Files.

## Supplemental Information

Supplemental information for this article can be found online at http://dx.doi.org/10.7717/peerj.7049#supplemental-information.

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
