# Peer review of "Influence of foetal inflammation on the development of meconium aspiration syndrome in term neonates with meconium-stained amniotic fluid"

_PeerJ, doi:10.7717/peerj.7049_

## Round 0.1 · original submission · Major Revisions

Dear Authors,
The Reviewers found your manuscript very interesting, however they are recommending an extensive revision in order to achieve publication.
I would suggest you take into consideration the Reviewers' comments, discuss and incorporate them within your manuscript in order to reach the standard requested for publication.
Best regards

Salvatore Andrea Mastrolia
PeerJ Academic Editor

Reviewer 1 ·

Basic reporting

Basically the manuscript is relatively well written, but the result section is rather too brief. For example, there are no mention about acute-phase inflammatory reaction score in the text, even though the score 3 seems to have statistical significance in the table. It is better to describe it in detail. Please see a reference below;
Hofer N, Jank K, Strenger V, Pansy J, Resch B. Inflammatory indices in meconium aspiration syndrome. Pediatr Pulmonol. 2016 Jun;51(6):601-6. doi: 10.1002/ppul.23349. Epub 2015 Dec 10. PubMed PMID: 26663621.

Experimental design

1. Line 53-54 “funistis and chorioamnionitis”: Based on their definition in the Methods section, the terms seem to mean acute inflammation in this study. Because there is another entity, chronic chorioamnionitis, which is not related to amniotic infection syndrome, the term “acute chorioamnionitis and funisitis” should be used through the whole text.
2. Line 117: the definition of term (neonate) is required, for example born at >=37 weeks of gestation.
3. Line 117-118 “Level II neonatal intensive care unit”: the “Level II NICU” is not familiar to non-pediatric people. Please describe it briefly. Is there any chance term neonates with meconium stained amniotic fluid do not admit to level II NICU? maybe the neonates with more severe symptoms only could be enrolled. If so, it should be described as another limitation.
4. Line 131-132 ”chorioamnionitis was defined as infiltration of neutrophils identified in chorionic plate…”: What is the reason the location of inflammation is limited to the “chorionic plate”? In general, acute chorioamnionitis is defined as neutrophil infiltration in the chorioamniotic membranes, even though acute inflammation in the chorionic plate is also considered as acute chorioamnionitis. Please confirm the diagnostic definition of acute chorioamnionitis from the pathologist of your institution.

Validity of the findings

Line 52-54: the authors said their aim was to investigate whether the incidence of MAS is associated with the presence of funisitis and chorioamnionitis, but the conclusion said some clinical variables (male sex, antenatal asphyxia) were also identified as independent variables. It is better to amend the aim or conclusion appropriately.

Additional comments

The results in this study supports the previous reports on MAS, however there are no definite novel findings. If clinical progression and/or treatment outcomes in detail are added in the results, readers would be more interested in the study.

Line 165 “Of 99 neonates...”: it is better to describe it as “ of 99 term neonates with meconium-stained fluid…”

Table footnote line 4: the details of the acute-phase inflammatory response score is not in “Table S1” but in the supplementary information. Why don’t you put it in the Method section?

·

Basic reporting

The manuscript has lots of grammatical errors which need to be corrected.
Line 49 - “ Although asphyxia” - Asphyxia is not the sole cause for meconium passage in utero and other factors such as in utero infections, placental insufficiency, post maturity, IUGR etc are involved in MSAF/ MAS. So the sentence could be rephrased as “fetal distress and subsequent…”
Line 50 - “stimulation have” - Should be changed to “stimulation has”
Line 51, 52 - “occurs in the absence of asphyxia, and the presence of additional causatives such as intraamniotic inflammation has been extrapolated” -change to “ occurs due to other pathologies such as intraamniotic inflammation ”
Line 57, 58 - “Dependence of MAS development on clinical variables and histopathological findings were assessed”. Modification - “ Clinical variables and histopathological findings associated with the incidence of MAS were studied”
Line 64, 65 - “In neonates with meconium stained amniotic fluid, male sex and funisitis, as well as evidence of antenatal asphyxia, were identified as independent variables for MAS development” - Modification - “In neonates with meconium stained amniotic fluid, male sex ,funisitis and low 1-minute APGAR score were identified as independent variables for MAS development”. 1 minute APGAR is not a definitive indicator of antenatal asphyxia and such an inference is not appropriate
Line 75 - 78 - “ In addition, advanced therapeutic regimens for severe respiratory failure using surfactant replacement, nitric oxide  inhalation, and extracorporeal membrane oxygenation, led to a marked decrease in mortality and morbidity associated with MAS” - Modify - The use of surfactant replacement, nitric oxide therapy and extracorporeal membrane oxygenation has lead to a marked decrease in mortality and morbidity associated with MAS in the recent times”
Line 79 - “ newborn born at a “ modify - “ newborn born in a”. “Even at a” - Modify - “ Even in a “
Line 81 - “ do not provide early, precise discrimination of neonates who are likely to develop severe respiratory failure, from their peers, who only experience transient respiratory failure”
Modify -“ do not provide any insights as to who are likely to develop severe respiratory failure ”
Line 141 - “In our unit, routine blood tests are scheduled at admission and on day 1 for neonates with meconium- stained amniotic fluid, where blood gas analysis” modify - “In our unit, routine blood tests that are done for neonates born through MSAF are blood gas analysis, cell counts, blood sugar level, and serum inflammatory markers and these tests were done in the study subjects”
Line 142 - 14 8 -“ For the inflammatory markers, elevation of C-reactive  protein (CRP;>0.3 mg/dL), α1-acid glycoprotein (α1AG; >20 mg/dL), and haptoglobin (>13 mg/dl) have been routinely assessed since 1983 using turbidimetric immunoassay (Quick Turbo, Shino-Test Corporation, Japan), based on previous reports, which found early diagnostic  property of these biomarkers and composite scores calculated from these values, or acute-phase  inflammatory reaction score, for severe neonatal infection (Speer et al., 1983; Ipek et al., 2010;  Nakamura et al., 2015) (see Article S1 for details of this composite score)”. modify - “ An acute-phase inflammatory reaction score predicting severe neonatal infection was calculated based on the values of CRP, Alpha1-acid glycoprotein and haptoglobin”
Line 151 - “ First,…” modify - no need for using first. Start the sentence with “ The clinical variables were …”
Line 152 - 154 - Needs to be changed. What the authors intend to say is not clear.
Line 168 - “statistically invariant” modify into - “statistically not significant”
Line 182 - 183 - “suggesting the importance of inflammatory and genetic factors in investigating the mechanism of MAS”. Modify - “ pointing towards the role of inflammatory and genetic factors in the pathophysiology of MAS”
Line 195 - “ additional control variable” - change to “ an additional factor involved in the development of MAS”
Line 229 - “ Instead, CRP, α1AG and haptoglobin were monitored at admission and on day 1, which are known to increase plasma IL- 1βand IL-6 levels.” This sentence is confusing. It is IL 6 secreted from the neutrophils that will result in the increase in CRP production in the liver”
232 - “Were not identified “ modify to “ were identified”
234 - “ at the timing of admission” - modify - “ at the time of admission”


- References - Needs major modification. Doi is usually not included in prescribed referencing standards such as Vancouver standards.

Experimental design

The incidence of MAS is very high in the study group. The authors need to state the possible reason for this. Was it because MAS was over diagnosed?
APGAR at 1 of <7 is not considered a low APGAR. It is not indicative of intrapartum asphyxia. Many guidelines such as ACOG take APGAR values that are much lower and at a later time as being significant.
The authors have specified “high birth weight” as associated with MAS. But the difference of birth weight between the groups is only 200 - 250 grams. Though it is statistically significant, what is the clinical significance of this?
Inflammatory markers are invariably elevated in MAS even without any underlying infection. Did authors look at the blood culture? That is the gold standard to diagnose sepsis.
What is the utility of histopathological funisitis? It will not help in predicting MAS earlier as HP Examination will take time.

Validity of the findings

Did the authors look at the requirement of resuscitation between the two groups? Along with cord blood gas, it would have been an important variable
The Wiswell definition is used to classify the severity of MAS. <40% FiO2 and less than 48 hours duration of respiratory support requirement comes under mild MAS. The authors have not included mild MAS here. Is there any specific reason why mild MAS was excluded?
Was thin vs Thick meconium studied?
What was the weight for gestational age status? - SGA/ IUGR Vs AGA. SGA s are a very important risk factor for MAS
Why preterms were excluded? MAS does occur in preterms and may be chorioamnionitis/ funisitis play a more important role in preterm with MAS than term with MAS.

Additional comments

The manuscript has a lot of grammatical errors and need to be revised.

---

## Round 0.2 · accepted · Accept

Dear Authors,
I would like to compliment with you for the efforts provided in addressing the Reviewers' comments.

The Reviewers felt that your manuscript has been considered suitable for publication and can be accepted in its current form.

Best regards

Salvatore Andrea Mastrolia
PeerJ Academic Editor

Reviewer 1 ·

Basic reporting

The manuscript is well-written with the appropriate responses for the reviewer's suggestions.

Experimental design

experimental design is OK with the clear description of the limitations.

Validity of the findings

The results are not quite novel, but they support the previous findings well.